# Evolution of heterogeneous perceptual limits and indifference in competitive foraging

**Richard P. Mann** [ORCID] *

Department of Statistics, School of Mathematics, University of Leeds, Leeds, United Kingdom

* r.p.mann@leeds.ac.uk

**Data Availability Statement:** All relevant data are within the manuscript and its Supporting information files.

**Funding:** This research was supported by a UK Research and Innovation Future Leaders

## Abstract

The collective behaviour of animal and human groups emerges from the individual decisions and actions of their constituent members. Recent research has revealed many ways in which the behaviour of groups can be influenced by differences amongst their constituent individuals. The existence of individual differences that have implications for collective behaviour raises important questions. How are these differences generated and maintained? Are individual differences driven by exogenous factors, or are they a response to the social dilemmas these groups face? Here I consider the classic case of patch selection by foraging agents under conditions of social competition. I introduce a multilevel model wherein the perceptual sensitivities of agents evolve in response to their foraging success or failure over repeated patch selections. This model reveals a bifurcation in the population, creating a class of agents with no perceptual sensitivity. These agents exploit the social environment to avoid the costs of accurate perception, relying on other agents to make fitness rewards insensitive to the choice of foraging patch. This provides a individual-based evolutionary basis for models incorporating perceptual limits that have been proposed to explain observed deviations from the Ideal Free Distribution (IFD) in empirical studies, while showing that the common assumption in such models that agents share identical sensory limits is likely false. Further analysis of the model shows how agents develop perceptual strategic niches in response to environmental variability. The emergence of agents insensitive to reward differences also has implications for societal resource allocation problems, including the use of financial and prediction markets as mechanisms for aggregating collective wisdom.

## Author summary

Competition for resources is a major determinant of behaviour in both humans and other animals, as individuals seek resources that are both plentiful and under exploited by others. In order to identify these opportunities, individuals require high-quality information about the world, via their senses or by costly research into the options available to them. The price of conducting such research or maintaining the sophisticated sensory apparatus required is a drain on the foraging returns the individual can acquire, and thus there is a trade off between the benefits of effective foraging and the costs involved. In simple

Fellowship MR/S032525/1 https://www.ukri.org/
The funders had no role in study design, data
collection and analysis, decision to publish, or
preparation of the manuscript.

**Competing interests:** The authors have declared
that no competing interests exist.

models of foraging behaviour, identical individuals that compete equally receive identical
rewards regardless of where they choose to forage, since any advantage between choices is
ultimately competed away. This presents an opportunity to make less effort in identifying
the best foraging opportunities and avoid paying the associated costs. Here I show how
this leads to the emergence of two distinct types of behavioural types: informed agents
that continue to assess the value of different options, and uninformed agents who choose
at random. Investigating further, I show how informed agents further diversify according
to the variety of options they are likely to encounter.

## Introduction

While many models of collective behaviour consider groups that are composed of identical
individuals (with identical motivations and rules of behaviour), much recent work in this field
has focused on the importance of heterogeneity: differences between individuals within a
group. Variation in the behaviour of individuals within groups has been associated empirically
with differences in personality [1–3], and group behaviour has been shown to depend on the
distribution of individual behavioural characteristics [4, 5]. Theoretical models have also
highlighted the importance of heterogeneity in producing efficient collective behaviour. For
example, bold individuals may lead groups to new resources while shy individuals provide col-
lective cohesion [1]; a similar effect is seen in models with informed and uninformed individu-
als respectively [6, 7]. Differences in individual motivations are an important source of
variation in the behaviour of groups composed of rational agents [8], while groups with mixed
evidence thresholds for decision making can make faster and more accurate decisions than
those composed of identical agents, [9]. Both theoretical and empirical evidence [10–14] has
highlighted the importance of 'noise' in collective behaviour. This noise may result from unob-
served differences between individuals that cause them to act in unpredictable ways in appar-
ently identical situations.

While heterogeneity is thus of recognised importance in understanding collective behav-
iour, relatively few theoretical studies have addressed how these differences between individu-
als are generated and maintained. Contingent differences in information may be one source of
variation (e.g. the position of an agent in its environment may lead to specific knowledge that
others lack). Other differences may be more fundamental consequences of adaptive processes.
For example studies of collective navigation [7, 15, 16] have demonstrated the evolution of dis-
tinct classes of agents: informed individuals, who specialise in obtaining accurate private navi-
gational information and who attend little to the movements of others, and uninformed
individuals, who conversely follow the movements of others in the group and expend few
resources on obtaining reliable private information. These studies reveal that heterogeneity
that is relevant for collective outcomes can arise from the collective dilemma itself. This raises
the question: what other social contexts permit analogous specialisation?

One important decision many animals must make within the competitive environment of a
social group is the choice of where to forage for food or other resources. A familiar case is the
large herds of many herbivore species that graze on grasslands, but competition for resources
within a social population is ubiquitous across animal taxa (see [17] for one review of many
examples). In each case, individuals must choose in a context where the rewards they receive
depend on the choices made by other agents. In ecology a standard model of competitive equi-
librium in such scenarios is the Ideal Free Distribution (IFD). The IFD [18, 19] describes how
'ideal' agents—ones with perfect knowledge of their environment—who are 'free'—experience

no cost of movement between different sites—will distribute themselves between resources of different quality. If each resource is entirely consumed by the individuals exploiting it, and if all individuals are identical and gain identical shares of their chosen resource, then the IFD predicts that the distribution of individuals will be proportional to the quality of the resources, a result known as 'input matching'. This distribution of agents is a Nash equilibrium [20], such that no individual can improve their expected reward by changing the patch they occupy, and in which all agents receive the same reward.

Although many studies have purported to show that real animal population distributions are well approximated by the IFD (e.g. [21–29]), reanalysis of the data from these studies has consistently shown lower exploitation of high quality resources than expected [17]. Experimental human studies also show below-expected exploitation of high quality resources [30]. This poses a problem for theoretical ecology—why do fewer individuals exploit high quality resources than expected? This 'under matching' is even more puzzling in social animals since collective decision-making typically creates a bias towards aggregation and even information cascades [31–34]. Milinski [35] correctly notes that there are many IFDs, depending on the nature of competition between agents, and asserts therefore that the IFD theory does not necessarily predict resource matching. Nonetheless, other researchers have suggested additional mechanisms to explain the perceived gap between theory and observation. Proposed mechanisms to explain under matching of high quality sites include costs of movement between patches, unequal competitive strengths of agents and perceptual limits that prevent agents from accurately knowing what resources are available [36, 37]. Addressing this last mechanism, models incorporating perceptual limits [38, 39] represent these as a fixed sensory limit imposed by the agents' physiology, which is typically assumed to be consistent across all individuals. It is the origin and distribution of these perceptual limits that I will investigate here.

In this paper I focus on the evolution of perceptual abilities in a competitive foraging environment. I follow the approach of ref. [7] by proposing a multilevel model for the evolution of perceptual abilities in a population of individuals competing for resources. In principle this population could composed of one or multiple species, but here I assume that agents are identical except in their perceptual ability. I analyse this model in a mathematically idealised form, considering the equilibrium of an infinite population, and also implement the model as an agent-based simulation. I show how the cost of perceptive abilities influences the distribution of these abilities in the population, and I demonstrate a bifurcation, with the emergence of a distinct class of perceptually-limited individuals. Furthermore I investigate how variability in the environment impacts on the distribution of perceptual ability and the consequences of the model for the utilisation of resources by the group as a whole.

## Materials and methods

I consider a two-level model of foraging and the evolution of perceptual limits, illustrated in Fig 1. At the lower level, individuals compete for resources, selecting foraging patches to optimise their individual share of available resources, and limited in their decision-making by an individual perceptual limit. At the upper level the population evolves by natural selection—individuals replicate according to their fitness, which is determined by the resources they are able to acquire in the lower-level game and the costs they pay for their perceptual abilities.

Below I first lay out the structure of the model, along with a mathematical analysis of the predicted equilibrium states, based on a dynamical system representing an idealised infinite population with complete time separation between the two game levels. I then describe an agent-based implementation of the model to test whether these predictions are robust for the case of finite populations with a mixing of time scales between the upper and lower level game.

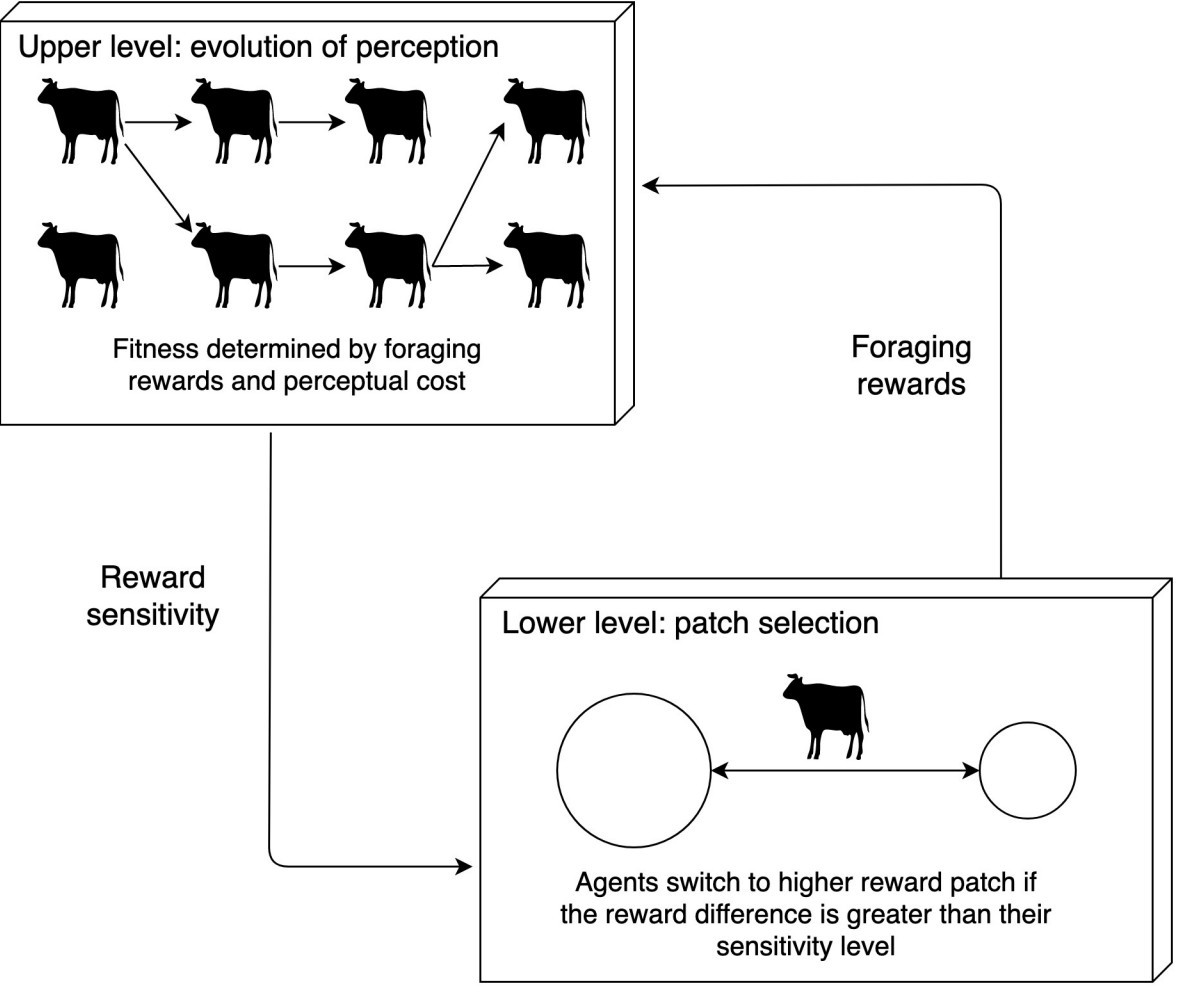

**Fig 1. Schematic of the interaction between patch selection and the evolution of sensitivity threshold as a multilevel game.** On the lower level game, agents select between patches with differing total resources. Each agent seeks to select the patch with greater reward, but is limited by their individual sensitivity threshold. In the upper-level game agents reproduce according to a fitness function determined by their patch-selection rewards and the cost of their sensitivity threshold.

### Lower-level game: Patch selection

I consider a population of individuals foraging across two patches, A and B. Without loss of generality one can assume that the total resources across the two patches are equal to one, and I specify that the ratio of the two patches is $r$, such that the resources on patch A are $1/(1 + r)$ and those on patch B are $r/(1 + r)$. Each individual is able to freely choose a patch on which to forage, and can change their patch without cost. Ultimately, each individual receives an equal share of the resources on the patch on which they are foraging when the game concludes.

Every individual has a perceptual sensitivity threshold, $\tau$, that can differ from one individual to another. This threshold dictates the conditions under which the individual will switch which patch it will forage on. Individuals can only distinguish reward differences between patches above their sensitivity threshold. If, with the current distribution of agents, the current patch has a reward at least $\tau$ less than the alternative, then the individual will switch. If the alternative patch has a reward less than $\tau$ greater than the current patch then the individual will remain on the current patch. For any given iteration of the lower level game I assume that

there is a fixed distribution of agents such that $\rho(\tau)$ is the probability density that a given agent has sensitivity threshold $\tau$.

Let $\rho(\tau, A)$ be the density of agents with sensitivity threshold $\tau$ on patch A. Similarly define $\rho(\tau, B)$, such that $\rho(\tau) = \rho(\tau, A) + \rho(\tau, B)$ is the density of all agents with sensitivity threshold $\tau$ across both patches.

For any given distribution, each individual on patch A will have an anticipated reward (assuming no further change) given by:

$$R_A = \frac{1}{1+r} \frac{1}{\int_0^\infty \rho(\tau, A) d\tau},$$

and similarly each individual on patch B anticipates a reward given by:

$$R_B = \frac{r}{1+r} \frac{1}{\int_0^\infty \rho(\tau, B) d\tau}.$$

Define the difference between these anticipated rewards, $\Delta R \equiv R_B - R_A$. On whichever of the two patches has the lower anticipated reward, those with a threshold $\tau \leq |\Delta R|$ switch to the more rewarding patch at rate one (defining the arbitrary timescale). Those agents with $\tau > |\Delta R|$ switch randomly between the two patches at a slower rate $\gamma$. For each $\tau$ then,

$$\frac{d\rho(\tau, A)}{dt} = -\frac{d\rho(\tau, B)}{dt}$$
$$= \begin{cases} \rho(\tau, B) H(-\Delta R) - \rho(\tau, A) H(\Delta R), & \forall \tau \leq |\Delta R| \\ \\ \gamma(\rho(\tau, B) - \rho(\tau, A)), & \forall \tau > |\Delta R| \end{cases} \tag{1}$$

where $H(\cdot)$ is the Heaviside step function.

In this implementation I assume that the game runs until equilibrium is reached, such that the distribution of agents between A and B is fixed for all values of $\tau$:

$$\frac{d\rho(\tau, A)}{dt} = 0 \ \forall \tau.$$

To derive the equilibrium distribution, first note that at equilibrium the reward to agents on patch B will be greater than or equal to that on patch A, i.e. there is an equilibrium reward difference $\Delta R^* \geq 0$. Any agent with sensitivity threshold $\tau$ below this equilibrium reward difference must be on patch B, since they would otherwise switch from patch A to B. Those agents with $\tau$ greater than $\Delta R$ should be equally distributed between patches A and B, since any difference in the distribution between A and B will be eroded by the random movements of these agents. I therefore define the proportion of agents on patch A as $x = 0.5 \int_{\Delta R^*}^\infty \rho(\tau) d\tau$, and identify the following condition for equilibrium, which one can solve numerically for any given distribution $\rho(\tau)$:

$$\Delta R^* = \frac{1}{1+r} \left( \frac{r}{1-x} - \frac{1}{x} \right) \tag{2}$$

## Upper-level game: Evolution of perception

For the purposes of mathematical analysis I assume a complete separation of time scales between the lower-level patch selection game and the evolution of perception. As such, I

attribute to each value of $\tau$ a reward based on the mean lower-level rewards of agents with that sensitivity threshold, as if an infinite number of lower level games had been played for any given step of evolution in perception. When only one reward ratio is possible, this implies that each value of $\tau$ is assigned a mean reward $\mathbb{E}(R)_\tau$:

$$
\begin{aligned}
\mathbb{E}(R)_\tau &= \frac{\rho(\tau, A)R_A + \rho(\tau, B)R_B}{\rho(\tau, A) + \rho(\tau, B)} \\
&= \begin{cases} \frac{1}{x(1+r)} + \Delta R^*, & \tau \le \Delta R* \\ \\ \frac{1}{x(1+r)} + 0.5\Delta R^*, & \tau > \Delta R*. \end{cases}
\end{aligned}
\tag{3}
$$

When multiple patch ratios are possible I weight the outcomes for each ratio by the proportion of times that ratio occurs.

Each individual also incurs a cost, $c(\tau)$ based on their sensitivity threshold, for which I specify the following principles:

1. Cost increases with perceptual sensitivity: $c(\tau') < c(\tau) \Leftrightarrow \tau' > \tau$

2. No perceptual sensitivity incurs no cost: $\lim_{\tau \to \infty} c(\tau) = 0$

3. Perfect perceptual sensitivity incurs a finite cost: $c(0) \neq \infty$

Based on these principles I model this cost via the function:

$$
c(\tau) = \frac{\alpha}{1 + \tau},
\tag{4}
$$

where $\alpha$ is a parameter allowing variable perception costs. The results in this paper are based on this cost function, however the model proposed here is general and other cost functions can be substituted. In particular, principle (iii) above could justifiably be removed, such that perfect perception incurs an infinite cost and is therefore not possible. I retain this condition to avoid making perceptual limits tautological. The cost function I have chosen is also convex, but the analytical results I will present do not rely on this property.

The upper-level game represents evolution of the sensitivity threshold in the population. The fitness, $F(\tau)$ of agents with a sensitivity threshold of $\tau$ is given by the mean reward for such agents in the lower-level game, minus the cost incurred for this level of perception.

$$
F(\tau) = \mathbb{E}(R)_\tau - c(\tau)
\tag{5}
$$

At the population level, sensitivity thresholds evolve according to the replicator equation:

$$
\frac{d}{dt}\rho(\tau) = \rho(\tau)\left(F(\tau) - \int_0^\infty \rho(\tau')F(\tau')d\tau'\right).
\tag{6}
$$

Equilibrium requires a steady state distribution such that $\frac{d}{dt}\rho(\tau) = 0$ for all $\tau$, and thus requires:

$$
\rho(\tau)F(\tau) = \rho(\tau)\int_0^\infty \rho(\tau')F(\tau')d\tau',
\tag{7}
$$

which is equivalent to stating that the overall reward, including perceptual cost, must be the same for all occupied ($\rho(\tau) > 0$) perceptual strategies.

**Equilibrium with one possible ratio.** To derive the equilibrium distribution of sensitivity thresholds, I first consider the case where there is only one possible patch ratio, $r$. For any

given distribution over sensitivity thresholds, $\rho(\tau)$, this implies a specific reward difference resulting from the lower-level patch selection game, as given by Eq 2, leading to the expected patch-selection rewards for each value of $\tau$ given by Eq 3. When the cost of perception is added to these patch-selection rewards, the overall reward for each value of $\tau$ is thus:

$$F(\tau) = \begin{cases} \frac{1}{x(1+r)} + \Delta R^* - c(\tau), & \tau \leq \Delta R* \\[2mm] \frac{1}{x(1+r)} + 0.5\Delta R^* - c(\tau), & \tau > \Delta R*. \end{cases} \tag{8}$$

It is required that the value of $F(\tau)$ be the same for any occupied strategy at equilibrium, such that the equilibrium condition in Eq 7 is met. Looking at each case in Eq 7 separately, one see that this condition cannot be true for multiple strategies with $\tau \leq \Delta R^*$, or for multiple strategies with $\tau > \Delta R^*$, since in each case $F(\tau)$ differs only through variation in the perceptual cost function $c(\tau)$, which decreases strictly monotonically with $\tau$. Any strategy with $\tau < \Delta R^*$ is always dominated by $\tau = \Delta R^*$, and any strategy with $\tau > \Delta R^*$ is dominated by $\tau = \infty$. Therefore, the only two possible stable strategies are $\tau = \Delta R^*$ and $\tau = \infty$. To determine the proportion of agents employing each strategy, I look for an equilibrium where the overall rewards for each are the same:

$$F(\infty) = F(\Delta R^*)$$

$$\Rightarrow c(\Delta R^*) = 0.5\Delta R^*, \tag{9}$$

where I have utilised the property that $c(\infty) = 0$. For my specific choice of cost function, this implies:

$$\Delta R^* = \frac{\sqrt{1 + 8\alpha^2} - 1}{2}. \tag{10}$$

Recall that the proportion of agents on patch A, $x$, is half the proportion of agents with $\tau > \Delta R^*$, and therefore $x = 0.5\rho(\infty)$. Having determined $\Delta R^*$, one can solve for $x$ and therefore $\rho(\infty)$ from Eq 2, giving:

$$\rho(\infty) = \frac{(\Delta R^* - 1) + \sqrt{((\Delta R^* - 1)^2 + 4\Delta R^*)}}{\Delta R^*}. \tag{11}$$

In cases where there is no solution for $\rho(\infty)$ below one, this indicates that the patch reward difference necessary to support the $\tau = \Delta R^*$ strategy is impossible to attain, and all agents thus converge to $\tau = \infty$.

The above analysis relies on the strict monotonicity of $c(\tau)$. This is in most cases a realistic assumption of real sensory costs. However, if instead $c(\tau)$ remains constant beyond some threshold, $\tau_{\text{plateau}}$, the argument can be modified as follows. If $\tau_{\text{plateau}}$ is above the equilibrium value of $\Delta R^*$ established above, then agents with $\tau > \Delta R^*$ continue to experience a selective pressure to increase $\tau$ for $\tau < \tau_{\text{plateau}}$; this maintains the bifurcation in strategies shown above. Above $\tau_{\text{plateau}}$ there is no change in perceptual cost, and thus no fitness gradient, so agents will diffuse equally among all values of $\tau > \tau_{\text{plateau}}$. If instead the value of $\tau_{\text{plateau}}$ is below $\Delta_R^*$ then there will be a selective pressure against agents with $\tau > \tau_{\text{plateau}}$, since these agents pay no lower a cost while potentially losing the ability to gain from discerning patch reward differences. Hence, all agents will converge to $\tau = \tau_{\text{plateau}}$. This is born out in agent-based simulations (described below), shown in S1 Fig.

**Equilibrium with multiple possible ratios.** For a given distribution of sensitivity thresholds $\rho(\tau)$, any given patch ratio leads to a specific reward difference between the higher and

lower quality patches at equilibrium, as given by Eq 2. As specified above, in any single patch-selection game, agents with a sensitivity threshold that is below this reward difference will all occupy the higher quality patch, while those with values above the reward difference will be equally spread across both patches. When there are multiple possible patch ratios, this creates the possibility for more perceptual strategies to coexist. To see how this works, consider the case of two possible patch ratios, $r_1$ and $r_2$, with $r_1 > r_2$. These patch ratios in turn give rise to equilibrium reward differences $\Delta R_1^*$ and $\Delta R_2^*$, with $\Delta R_1^* > \Delta R_2^*$. A strategy with $\tau > \Delta R_1^*$ will always choose a random patch in all patch selection games, and thus is dominated by a strategy of $\tau = \infty$, which also does so with lower perceptual cost. A strategy with $\Delta R_2^* < \tau < \Delta R_1^*$ chooses the higher quality patch when the patch ratio is $r_1$, but chooses randomly when the patch ratio is $r_2$, and thus is dominated by a strategy of $\tau = \Delta R_1^*$. Finally, any strategy with $\tau < \Delta R_2^*$ always chooses the higher quality patch, but is dominated by a strategy of $\tau = \Delta R_2^*$, which does the same with lower perceptual cost. Therefore three distinct equilibrium strategies are possible: (i) $\tau = \infty$; (ii) $\tau = \Delta R_1^*$; and (iii) $\tau = \Delta R_2^*$. Extending this argument shows that for $n$ distinct possible reward ratios, $r_1 > r_2 > \ldots > r_n$, there are $n + 1$ corresponding possible perceptual strategies.

As with the case of one patch ratio, the specific values of $\Delta R_1^*, \ldots, \Delta R_n^*$ and the proportion of agents adopting each perceptual strategy can be derived by equating the overall rewards for all occupied strategies at equilibrium. This can be done recursively as follows. First consider the two strategies $\tau = \infty$ and $\tau = \Delta R_1^*$. For all patch ratios other than $r_1$, these two strategies lead to the same outcome. If patch ratio $r_i$ occurs with a relative frequency $f_i$ then equating the overall rewards gives:

$$
\begin{aligned}
F(\infty) - F(\Delta R_1^*) \quad &= 0 \\
&= c(\Delta R_1^*) - 0.5 f_1 \Delta R_1^*
\end{aligned}
\tag{12}
$$

This equation leads to a quadratic solution $\Delta R_1^*$ in the case of my chosen cost function (as in Eq 10), or can be solved numerically for arbitrary cost functions $c(\tau)$.

Now consider strategies $\tau = \Delta R_1^*$ and $\tau = \Delta R_2^*$. Agents adopting these strategies accrue the same patch-selection rewards except when the patch ratio is $r_2$. Equating the overall rewards gives:

$$
\begin{aligned}
F(\Delta R_1^*) - F(\Delta R_2^*) \quad &= 0 \\
&= -c(\Delta R_1^*) + c(\Delta R_2^*) - 0.5 f_2 \Delta R_2^* \\
&= c(\Delta R_2^*) - 0.5 f_1 \Delta R_1^* - 0.5 f_2 \Delta R_2^*.
\end{aligned}
\tag{13}
$$

Following this recursive logic, the general formula specifying all sensitivity thresholds is:

$$
c(\Delta R_k^*) = 0.5 \sum_{i=1}^{k} f_i \Delta R_i^*
\tag{14}
$$

The proportion of agents adopting each strategy can similarly be calculated recursively. For any given patch ratio $r_k$, the proportion of agents occupying strategies $\tau \leq \Delta R_k^*$ must be such as to produce the associated patch reward difference $\Delta R_k^*$, as given by Eq 2. This gives the

relation below:

$$\frac{1}{1 + r_k}\left(\frac{r_k}{1 - 0.5\sum_{j=k}^{n}\rho(\Delta R_j^*)} - \frac{1}{0.5\sum_{j=k}^{n}\rho(\Delta R_j^*)}\right) = \Delta R_k^* \qquad (15)$$

Given $\Delta R_n^*$ from above, this can be solved recursively starting with $k = n$ and proceeding to $k = 1$. For any $k$, if no valid solution exists for $\rho(\Delta R_j^*)$ between 0 and 1 then the population cannot support this strategy and therefore $\rho(\Delta R_j^*) = 0$

## Agent-based model

To test the validity of the mathematical equilibrium analysis above to dynamic and finite populations, I implemented an agent-based model of the two-level game. In this model a finite number of distinct agents, each with an individual sensitivity threshold, select between two sites. As above, the agents are free to change the patch on which they wish to forage freely and at no cost. In this implementation I identify all agents that would wish to change patch, based on the rewards currently available and their sensitivity threshold. Of those that would wish to change, one is selected at random to switch and the process is repeated until an equilibrium is reached where no agent wishes to change patches.

Each agent then subsequently receives an individual reward based on the patch they chose and the number of other agents selecting that patch in analogous fashion to the model above. Each agent also pays a cost based on their sensitivity threshold, and the sum of this reward and cost results in their fitness. Agents then reproduce via roulette reproduction with mutation of the sensitivity threshold, with all agents in the previous generation being removed and the population size remaining constant. Since it is expected that at least some agents' sensitivity threshold will diverge towards $\tau \to \infty$, I map and mutate sensitivity thresholds on a logistic scale: $logistic(\tau) = 1/(1 + exp(-\tau))$. I assume normally distributed mutations on this scale, such that:

$$logistic(\tau_{child}) \sim \mathcal{N}(logistic(\tau_{parent}), \sigma), \qquad (16)$$

where $\sigma$ is the scale of mutation. The new population of agents then plays the lower level game and the process is repeated. Over each iteration of the population reproduction a record is kept of the distribution of sensitivity thresholds, so as to reveal the dynamics of the strategy evolution in addition to the equilibrium state. Code for running agent-based models and reproducing the results in this paper is included in S1 Code.

## Results

### Strategy bifurcation

I first considered a population which repeatedly faces a simple foraging scenario: to select between two patches with differing resources, such that the richer patch always contains a multiple $r$ of the resources in the poorer patch. Agents are initially assigned to a random patch and switch based on the relative rewards available at each and their individual sensitivity thresholds, according to the model described above. Once the distribution of the population reaches equilibrium, rewards are allocated (including the cost of perception) and the sensitivity thresholds of the agents evolves according to the second tier of the model, before the lower level game is repeated. I initialised the population of 1000 agents such that all individuals begin with perfect perceptual sensitivity ($\tau = 0$), and tracked the evolution of this parameter in the

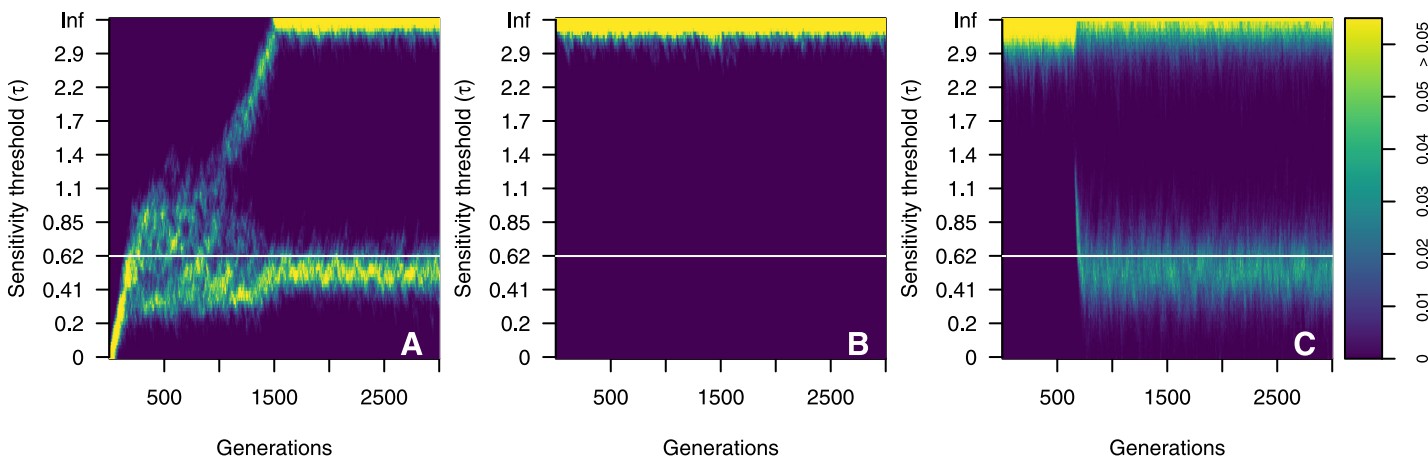

**Fig 2. Evolution of agents' sensitivity threshold over 3000 generations, showing the proportion of agents across potential perceptual strategies (in 100 logistically-spaced intervals).** In panel A the population evolves from an initially homogeneous population with $\tau = 0$, showing a bifurcation into two distinct sub-populations. Simulation parameters: 1000 agents, $\alpha = 0.5$, $r = 5$, $\sigma = 0.0025$. The white line indicates the theoretical expected sensitivity for agents with finite sensitivity threshold. In panel B the population is initially homogeneous with $\tau = \infty$, with the other simulation parameters unchanged. Here the population remains distributed close to the initially homogeneous starting distribution. In panel C the mutation rate is increased by a factor of three ($\sigma = 0.0075$). The population initially all remain close to $\tau = \infty$, before a rapid bifurcation in part of the population moves to the more sensitive perceptual state.

population over time. Fig 2A shows how the proportion of agents with differing sensitivities changes over 3000 generations, in a scenario where $r = 5$ and $\alpha = 0.5$, and with a mutation parameter $\sigma = 0.0025$. This shows that the population diverges from initial state of perfect perception with all agents losing perceptual sensitivity. After approximately 100 generations, when agents are clustered around a sensitivity of $\tau \simeq 0.4$, the population bifurcates into two distinct subgroups. One subgroup reaches a stable sensitivity threshold (in this case $\tau \simeq 0.5$). The second subgroup continually continues to lose perceptual ability until $\tau \to \infty$, becoming completely indifferent to the available rewards on each patch. Indicated by a white line is the predicted sensitivity threshold based on these simulation parameters from Eq 10. The agents retaining a finite sensitivity threshold tend to be clustered just beneath this level (i.e. have somewhat more sensitive perception), a result of there being greater costs to being just above the critical threshold than just below it.

I repeated the simulation process with the same parameters as above, but this with all agents initiated to have no perceptual sensitivity ($\tau = \infty$). In this case, shown Fig 2B, the agents remained clustered around the initial state without significant variation over 3000 generations. This reflects the relative difficulty of gaining, rather than losing perceptual sensitivity. An agent that mutates to have slightly more sensitivity gains nothing until that sensitivity reaches a point where $\tau$ is below the difference in reward between the two patches. In the case where $r = 5$ and all other agents are evenly spread between the two patches, that threshold occurs at $\tau = 2(5/6 - 1/6) = 1.33$. Increasing the mutation rate by a factor of three to $\sigma = 0.0075$ allows the agents to reach a mixed equilibrium, with some agents occupying the perceptive strategy; the observed bifurcation in this case is very rapid and a occurs when some of the agents reach $\tau \simeq 1.33$, as shown in Fig 2C. This suggests that transitions away from initially homogeneous and unperceptive populations, in which agents gain new perceptual abilities, are likely to appear sudden and occur preferentially when mutation rates are high, whereas the loss of perceptual sensitivity can occur gradually and under lower mutation rates.

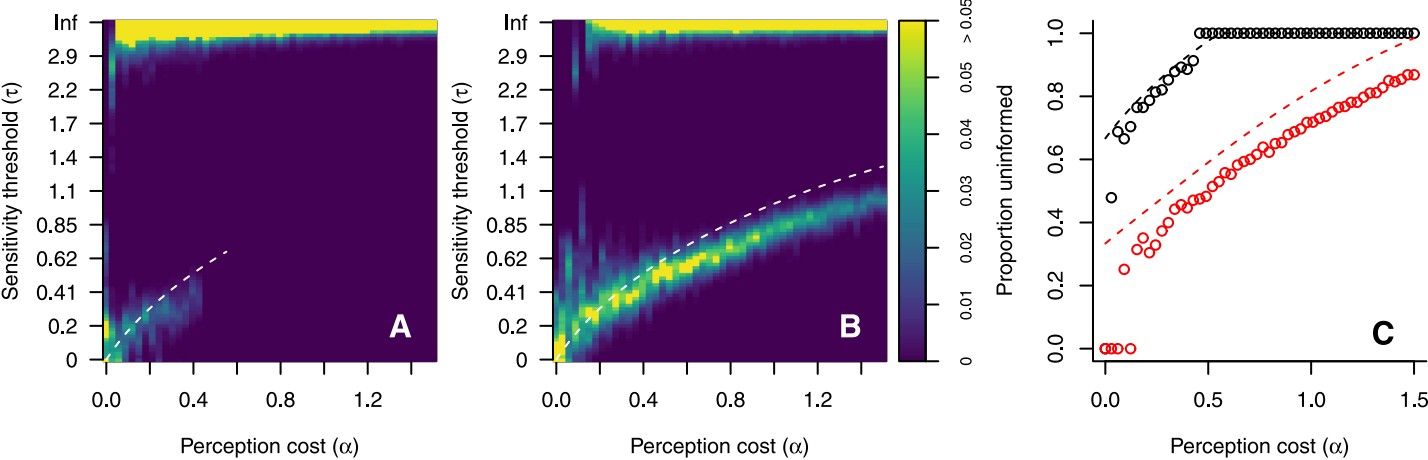

**Fig 3. Stable distribution of sensitivity threshold ($\tau$) in simulations of the agent-based model for different values of perceptual cost $\alpha$, with $r = 2$ (A) and $r = 5$ (B).** In each case the white dashed line indicates the value of $\tau$ for informed agents predicted by the analytical mathematical model. (C) Proportion of uninformed individuals ($\tau > 2$) in the agent-based model (solid lines) versus the analytical prediction (dashed lines) for $r = 2$ (black) and $r = 5$ (red).

## Effect of patch ratio and perception cost on stable strategy

Having established the existence of a bifurcation in perceptual strategies, I tested how general this phenomenon is across differing patch ratios and perception costs. For patch ratios of $r = 2$ and $r = 5$ I ran evolutionary simulations as above for a range of perception costs $0 < \alpha < 1$, and evaluated the equilibrium distribution of agent sensitivity thresholds. I also calculated the proportion of 'uninformed' individuals, which here are taken to be those with $\tau > 2$ (from the long-run population in the agent-based model)—this threshold is chosen as a clear visual separator of the different sub-populations. The results of these tests are shown in Fig 3. These results show several key features. First, there exist three distinct regimes based on perceptual cost: (i) a low cost regime, where all agents have a low sensitivity threshold; (ii) an intermediate regime, where informed agents coexist with uninformed agents. In this regime the proportion of uninformed agents steadily rises as the cost of perception increases; and (iii) a high cost regime where all agents are uninformed. Secondly, the range of costs over which coexistence is stable depends on the patch resource ratio; higher ratios can support informed agents at higher perception costs. Thirdly, informed agents do not retain perfect perception, but become less sensitive to reward differences as the cost of perception increases. The decrease in perceptual sensitivity among informed agents and the proportion of uninformed agents in the agent-based model closely match the predictions of the analytical mathematical treatment above, with two caveats. First, for very low perceptual costs the number of uninformed agents is very low or zero. This is the result of mutation dynamics that create a natural spread of sensitivity thresholds among informed agents, which in turn creates a consistent benefit to retaining perceptual sensitivity when costs are very low. Secondly, as seen in the bifurcation example above, the steady-state sensitivity threshold of informed agents in the agent-based model is slightly stronger (i.e. lower $\tau$) than predicted. This is again the result of diffusion of strategies due to mutation: the cost of mutating to a strategy with $\tau$ slightly above the predicted value is greater than an equivalent mutation slightly below. This asymmetry tends to maintain the population average at a lower value such that mutations above the predicted value of $\tau$ are rare.

## Variable environments

The analyses above have considered patch selection where the ratio of resources between patches is fixed. In reality, patch resource ratios typically vary, and the perceptual sensitivity of a foraging agent must adapt to this variability. To illustrate the effect of patch ratio variation I employed an agent-based model as above, but instead of repeatedly encountering a fixed patch ratio in each foraging task, agents were presented with one of two different patch ratios representing low or high resource differences ($r = 2$ and $r = 10$ respectively).

Fig 4 shows an example of the evolution of sensitivity thresholds among a population of agents in this scenario, with perceptual cost $\alpha = 0.5$. The population is initially homogeneous with all agents having perfect perception ($\tau = 0$). After an initial period of approximately 100 generations, during which time the agents lose perceptual sensitivity but remain relatively homogeneous, the population then branches into three distinct groups. One group loses all perceptual sensitivity ($\tau \rightarrow \infty$), as in the case of one patch ratio above. Those agents that retain some perceptual sensitivity further bifurcate into two groups, each specialised to a different patch ratio. As above, the stable distribution of values of $\tau$ for each subgroup remains slightly below the theoretical prediction from the analytical model, shown by the white lines.

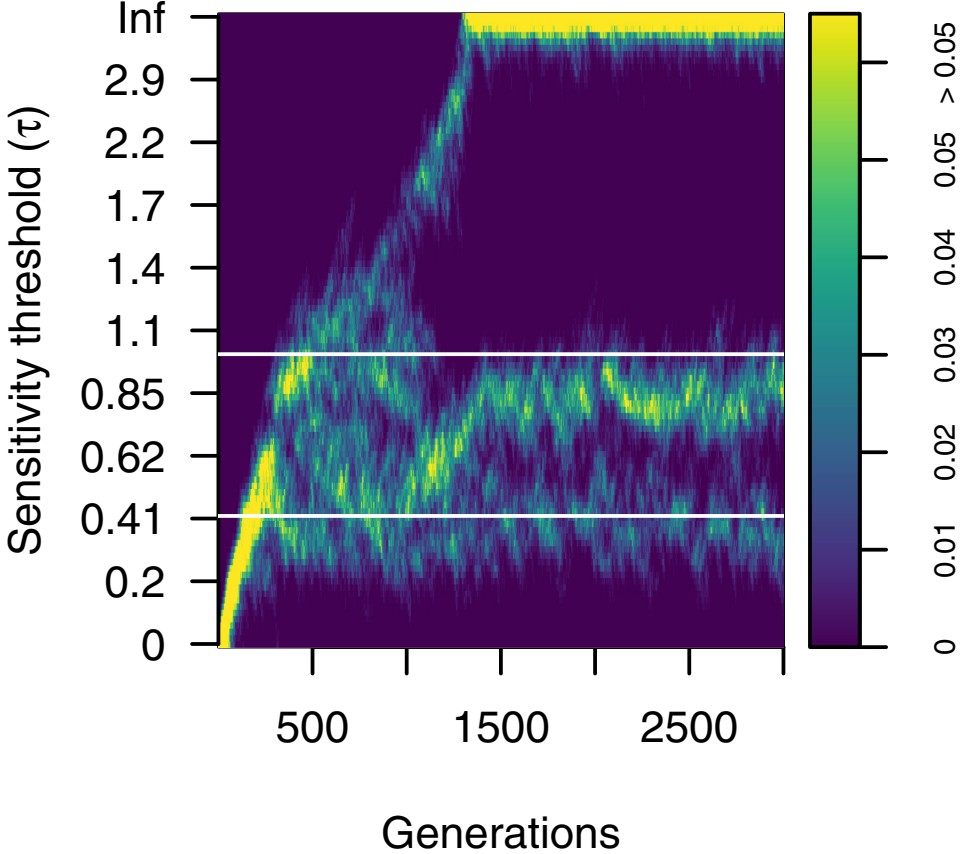

**Fig 4. Evolution of agents' sensitivity threshold over 3000 generations in the presence of variable patch ratio, showing the proportion of agents across potential perceptual strategies.** The population evolves from an initially homogeneous population with $\tau = 0$, and develops into three distinct sub-populations. Simulation parameters: 1000 agents, $\alpha = 0.5$, $r = 2$ and $r = 10$. The white line indicates the theoretical expected sensitivities for agents with finite sensitivity threshold.

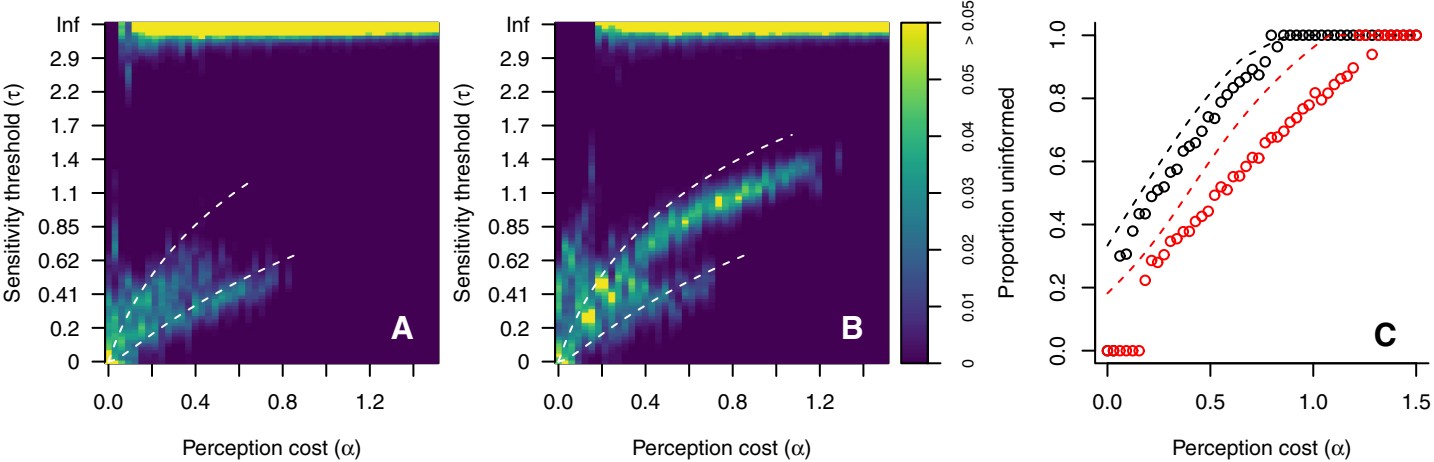

**Fig 5. Stable distribution of sensitivity threshold ($\tau$) in simulations of the agent-based model for different values of perceptual cost $\alpha$ and with variable patch ratios: (A) $r = 2$ and $r = 5$ and (B) $r = 2$ and $r = 10$.** In each case the white dashed line indicates the values of $\tau$ for informed agents predicted by the analytical mathematical model. (C) Proportion of uninformed individuals ($\tau > 2$) in the agent-based model (points) versus the analytical prediction (dashed lines) for $r = 2$ and $r = 5$ (black) and $r = 2$ and $r = 10$ (red).

Varying the cost of perception ($\alpha$) and the possible patch ratios ($r$) changes the stable distribution of agents across the three possible perceptual sensitivities. This is illustrated in Fig 5, which shows the steady state distribution of $\tau$ amongst agents in an agent-based model in a range of scenarios. Panel A shows the variation of agents sensitivity thresholds with varying perceptual cost when the possible patch ratios are $r = 2$ and $r = 5$. Panel B shows the equivalent variation when the possible patch ratios are $r = 2$ and $r = 10$. In each case the white lines show the predictions of the mathematical model. Panel C shows the proportion of 'uninformed' agents in each case, alongside the model predictions. As predicted by the mathematical analysis, there are 3 possible subgroups of agents, and the value of $\tau$ for these subgroups depends on the cost of perception but not on the patch ratios. However, the proportion of agents occupying each subgroup does strongly depend on possible patch ratios. Environments that typically contain high patch ratios (e.g. $r = 10$, panel B) sustain a larger proportion of informed agents, who are concentrated in the less perceptually sensitive (higher $\tau$) of the two 'informed' subgroups. These agents pay a relatively low perceptual cost and can take advantage of the large reward differences available when the patch ratio is high. Reducing the higher patch ratio to $r = 5$ (panel A) reduces the predicted range of $\alpha$ for which this less perceptive population exists, and almost eliminates agents of this type in the agent-based simulation. Conversely, the value of the higher patch ratio has no impact on the proportion of agents adapted to the highest perceptual sensitivity (lowest $\tau$) when the lower patch ratio remains unchanged, as predicted by Eq 15.

The mathematical analysis of equilibrium sensitivity distribution predicts that environments with many patch ratios should permit a greater number of possible different perceptual strategies, with predicted sensitivity values given by the solutions to Eq 14. Since patch qualities in real foraging contexts are generally highly variable and occupy a continuous range of values rather than being stereotyped into discrete values, it is natural to ask what distribution of perceptual sensitivity is expected under such conditions. Although many possible distributions of patch ratios are possible, a reasonable choice in the absence of context-specific details is the following: consider all the resources available on both patches as a unit interval, and divide that interval uniformly at random. Defining $r$ to be the ratio of the larger resulting piece to the smaller, this means that $r/(1 + r)$ is uniformly distributed between 0.5 and 1; equivalently $r$ is

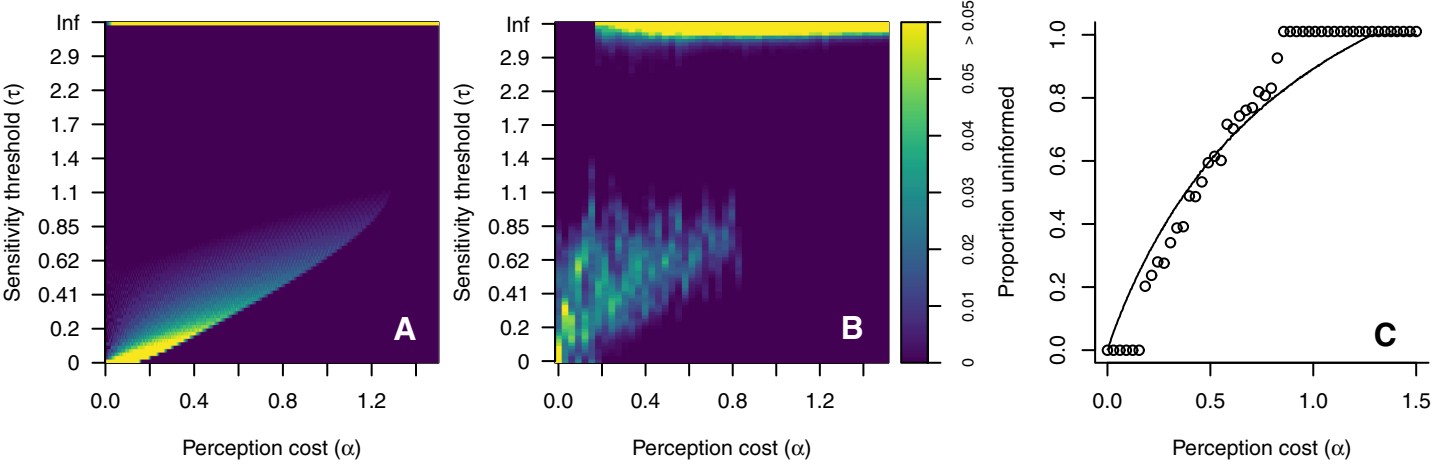

**Fig 6. Predicted and simulated stable distribution of sensitivity threshold ($\tau$) for widely-varying patch ratios ($r/(1+r) \sim U(0.5, 1)$): (A)** analytically-predicted distribution of stable sensitivity thresholds, colour-coded by expected occupancy; **(B)** steady-state distribution of sensitivities in agent-based simulations; and **(C)** proportion of uninformed individuals ($\tau > 2$) in the agent-based model (points) versus the analytical prediction (dashed lines).

distributed as $p(r) = 2/(1+r)^2$, for $r \geq 1$. To explore the consequences of this type of environment, I first chose 100 uniformly spaced values of $r/(1+r)$ in the interval from 0.5 to 1, and calculated the predicted equilibrium sensitivity values (Eq 14) and the corresponding proportion of agents predicted to occupy each sensitivity state (Eq 15). The resulting predictions, illustrated in Fig 6A, show a band of possible sensitivities that varies with perceptual cost: for very low values of $\alpha$ agents are predicted to cluster close to $\tau = 0$; the mean sensitivity and the range of sensitivity values increases as $\alpha$ increases, before tapering off for values of $\alpha$ above approximately 1.2, after which all agents are predicted occupy the uninformed state $\tau = \infty$. Next I simulated an agent-based model in which $r/(1+r)$ took the same 100 distinct values between 0.5 and 1, for the same range of perceptual cost values. The resulting distribution of agents shows qualitative similarities to the predictions in panel A, notably a broad range of sensitivity values amongst agents for intermediate values of $\alpha$, and all agents occupying the uninformed state for high values of $\alpha$. The proportion of uninformed agents in the agent-based model closely fits to the predicted value (panel C). A notable difference between the prediction and agent-based simulation results is seen for low values of $\alpha$: while agents are predicted to cluster close to $\tau = 0$, in the model they are widely spread for all but the very lowest values of $\alpha$. At the same time, fewer agents than expected occupy the uninformed state. This is similar to the pattern seen in Figs 3 and 5, and likely results from the natural spread of agents beyond the mathematical predictions due to mutation, which then opens exploitable opportunities for agents that would otherwise become uninformed to instead maintain a poor but finite perceptual sensitivity (high $\tau$).

## Discussion

The model I presented in this paper showed that agents evolve different sensitivity thresholds depending on the cost of sensory precision and the typical relative qualities of alternative patches. In particular, I showed that over a substantial range of the sensory cost parameter agents develop heterogeneous perceptual abilities, with a fundamental bifurcation of the population into 'informed' individuals who retain some discernment of the expected rewards at each patch, and 'uninformed' individuals who choose a patch at random. Informed individuals remain responsive to differences in expected rewards between two patches, but also decrease

their sensory precision as sensory cost increases. The proportion of uninformed individuals increases as the cost of sensory perception increases, and for high enough sensory costs no informed agents remain. Conversely, the greater the typical difference between the patches, the greater the proportion of informed individuals for any given sensory cost. In variable environments further heterogeneity emerges, as informed agents evolve a range of perceptual abilities, corresponding to different niches—each agent is adapted to benefit from a specific resource ratio. While I have investigated adaptation in an evolutionary framework, adaptation through agent learning would likely follow a similar pattern, owing to the close connection between evolutionary and learning dynamics in multi-agent systems [40–43].

A consequence of the population bifurcation is that the population as a whole does not distribute itself between two patches as predicted by the most-commonly described Ideal Free Distribution (one in which agents distribute themselves in proportion to patch quality), but instead 'under matches'. That is, the higher quality patch attracts a smaller proportion of agents than expected. This provides a competitive advantage to the informed individuals, which balances the cost of sensory perception at equilibrium. While sensory limitations have been posited as an explanation for under matching between populations and resources observed in real animal populations [38, 39], models that incorporate sensory limitations have typically imposed these as a fundamental facet of the population, and have assumed that these are identical among agents. Here I have shown how perceptual limitations can evolve within a population when perception carries costs. Focusing on the evolutionary origins of these limitations has shown that an assumption of identical sensory abilities is likely to be wrong. Nor are sensory limits likely to be broadly distributed; there is an fitness cost to maintaining any perceptual ability that is either insufficiently precise to distinguish between typical scales of reward imbalance the animal experiences, or to a one which is substantially more precise than required to do so. As a result, agents will tend to cluster around the sensitivity threshold necessary to take advantage of the typical resource ratios in the environment they habitually encounter or lose all sensitivity to patch rewards. A consequence of the bifurcation (and potentially further specialisation) of sensitivity thresholds in the population is that measured sensitivities in real agents, whether as individuals or as taken as an average, may not accurately predict collective behaviour if those agents are assumed to be distributed around a single central mode.

This model is not intended to be a unique or complete explanation for observed under matching. As noted by Milinski [35], there are many IFDs, and resource matching occurs only under particular assumptions about the nature of competition between agents—specifically that agents divide all the available resources on a patch equally. Moreover, alternative mechanisms such as movement costs (e.g. [44]) may also contribute to observed deviations from simple IFD models. It is likely that each empirical case study involves a mixture of these factors. Nonetheless, this model shows perceptual limits, which have been evoked as a mechanism to explain under matching, will evolve wherever perceptive accuracy carries a cost. Moreover, the model presented here makes two key qualitative predictions concerning the evolution and distribution of perceptual abilities: (i) such abilities should be distributed more unequally in populations exposed to highly variable environments, where patch ratios can take many (and possibly extreme) values; (ii) the existence of agents who possess or exercise essentially no perceptive abilities in social foraging contexts, for instance by never moving or selecting randomly between patches (i.e. patch selection being uncorrelated to patch quality or the current position of other individuals). As a corollary to this second prediction, the results shown in Fig 2 suggest that the evolution of new perceptual abilities that are relevant to social foraging is likely to occur preferentially when mutation rates are higher; it should therefore be expected to

occur relatively suddenly after a potentially-long period of stasis, whereas perceptual loss can occur more gradually and systematically.

The results here resemble those seen in models of evolving migratory strategies in collective navigation [7, 15, 16], where individuals within a flock bifurcated into two subgroups, again labelled 'informed' and 'uninformed', with informed individuals attending closely to the desired direction of travel (assumed to demand costly sensory abilities) and uninformed individuals attending to and following the movements of other group members. Whereas in those examples uninformed individuals relied on following informed peers when choosing where to move to reach the desired destination, in this model uninformed individuals rely on their informed peers to make their decision irrelevant, by equalising rewards for everyone. A similar phenomenon is seen in another recent model of foraging behaviour wherein high movement costs induce a subset of foragers to remain sedentary at the cost of poorer foraging returns [44]. This type of outcome is likely to apply across a wide range of game theoretic scenarios where rewards are equal across strategies at equilibrium; in these cases there is always an opportunity to avoid paying the cost of making an informed choice if one assumes that the system is at equilibrium, and thus that the rewards associated with all choices are the same. Examples include the Minority Game [45] and Rock-Paper-Scissors [46] as well as the patch selection game considered here. My results suggest that in all of these examples there will be a tendency for populations to split into clearly defined informed and uninformed subgroups. How many agents will choose or evolve to make uninformed decisions in such an environment depends on the cost of being informed, and the typical differences in payoffs between options.

Environmental variability produces greater diversification and specialisation among agents. Depending on the cost of perception, agents diversify into subgroups specialised to exploit each possible patch resource ratio. When the range of possible resource ratios is higher, this results in a greater range of perceptual costs that could support different specialisations (see Fig 5). I expect that similar specialisation would occur in the previously-studied collective navigational scenario [7, 15, 16] and in other comparable contexts if variation in rewards were introduced, for example by exposing a population to both critical navigation tasks (such as large scale migrations, that carry large penalties for failure) and less costly navigation tasks (such as local exploration for food). Environmental variation in resource types and availability provides a well-established ecological basis for generating specialisation in the form of differing individual preferences or foraging strategies, enabling different agents to exploit different resources [47, 48]. My results show that even where agents continue to exploit the same resources, specialisation in perceptual ability is likely to evolve to exploit variation in resource density. An interesting case of such specialisation might be represented by the evolution of varying sensory apparatus and associated nervous systems in early animals [49], in response to resource heterogeneity generated by the presence of the sessile 'Ediacaran Fauna' in a previously homogeneous environment [50, 51]. Notably, the results presented in this paper suggest that such gains in perceptual abilities are likely to be observed as a rapid transition, since large mutations are needed to overcome the fitness gradient between the uninformed state and the stable perceptive state.

The evolution of differing perceptual strategies, and in particular the emergence of uninformed agents, has implications for important collective behaviours in human groups. The Efficient Market Hypothesis (EMH) [52], which states that the actions of stockmarket traders act to encode all information in the price of the stock, and therefore that price movements are effectively random, can be seen as a close analogue to the IFD assumption in animal foraging. In this environment there is a clear example of the 'uninformed' agent: a 'passive investor' who invest in a representative set of all shares (for example, through an index tracker fund), rather

than attempting to select the stocks with the best expected rewards [53]. Only if a sufficiently high proportion of traders remain 'informed' will the market continue to allocate capital efficiently (in the sense of the EMH). This implies that if the cost of being informed is too high that markets could become dysfunctional as the proportion of passive investors increases. If the market experiences a sustained period in which the quality of different investments is relatively consistent, more participants may become 'uninformed', making the market as a whole less able to adapt to a new regime when the quality of investments becomes more diverse. Similar problems are faced in other competitive resource allocation problems, such as choosing which route to drive at which time to avoid traffic congestion [54], or the choice of which queue to join at the supermarket [55]. In all these cases one should consider that any assumption that agents allocate themselves in an 'efficient' or 'ideal' configuration is only valid if the cost of information is sufficiently small, and the stakes are sufficiently high. Taking the above argument in reverse, if one wishes to use the choices of agents as representative of some underlying ratio of resources, it is necessary to ensure that the costs of information are low, such that a substantial population of informed individuals can be maintained. This requirement should be borne in mind when interpreting the values of financial or prediction markets [56] as indicators of collective wisdom about future outcomes.

## Supporting information

**S1 Fig. Illustration of the equilibrium states reached when the cost function does not decrease beyond a threshold.** In each panel the patch ratio is $r = 5$, and the mutation parameter $\sigma = 0.0025$. The cost function remains constant for values of $\tau$ above a threshold (indicated by the red dashed line); panel A shows a threshold of 0.5, and panel B a threshold of 1.5. The white dashed line indicates the predicted equilibrium state in the absence of the threshold. When the threshold is below the expected equilibrium value for informed agents, all agents converge to this threshold. When the threshold value is above the expected equilibrium, informed agents converge to a lower equilibrium, while uninformed agents are distributed widely above the threshold.
(EPS)

**S1 Code. R code for reproducing results.** This code contains functions for reproducing Figs 2–6 and S1 Fig, along with all the code used to carry out the analyses in this paper.
(R)

## Acknowledgments

My thanks to David Leslie for informative discussions. Giacomo Baldo, Wolfram Barfuss, Graham Budd and Viktoria Spaiser gave helpful feedback on the manuscript.

## Author Contributions

**Conceptualization:** Richard P. Mann.

**Formal analysis:** Richard P. Mann.

**Funding acquisition:** Richard P. Mann.

**Investigation:** Richard P. Mann.

**Methodology:** Richard P. Mann.

**Project administration:** Richard P. Mann.

**Resources:** Richard P. Mann.

**Software:** Richard P. Mann.

**Validation:** Richard P. Mann.

**Visualization:** Richard P. Mann.

**Writing – original draft:** Richard P. Mann.

**Writing – review & editing:** Richard P. Mann.

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
