## [Decision Letter · Decision Letter 0]

13 Oct 2020

Dear Dr. Mann,

Thank you very much for submitting your manuscript "Evolution of heterogeneous perceptual limits and indifference in competitive foraging" for consideration at PLOS Computational Biology.

As with all papers reviewed by the journal, your manuscript was reviewed by members of the editorial board and by several independent reviewers. In light of the reviews (below this email), we would like to invite the resubmission of a significantly-revised version that takes into account the reviewers' comments.

The three reviewers were quite positive about the manuscript but have a number of concerns that need to be addressed. In particular, please attend to concerns about the generality of the results, extensions to variable environments, and competing mechanisms. Please provide a point by point response to all of the comments of each of the reviewers. Thank you.

We cannot make any decision about publication until we have seen the revised manuscript and your response to the reviewers' comments. Your revised manuscript is also likely to be sent to reviewers for further evaluation.

Sincerely,

Jessica C. Flack

Associate Editor

PLOS Computational Biology

Stefano Allesina

Deputy Editor

PLOS Computational Biology

The three reviewers were quite positive about the manuscript but have a number of concerns that need to be addressed. In particular, please attend to concerns about the generality of the results, extensions to variable environments, and competing mechanisms. Please provide a point by point response to all of the comments of each of the reviewers. Thank you.

Reviewer's Responses to Questions

**Comments to the Authors:**

Reviewer #1: The author discusses in his paper the evolution of perceptual limits allowing agents to distinguish patches of different quality. The author introduces a simple but rather generic mathematical model, discusses and analyzes it first analytically in the equilibrium case, and finally performs some agent-based simulations to test the predictions of the theory. A core results is the emergence of evolutionary branching for certain parameter ranges, where a co-existence of agents with no perceptual sensitivity and agents with perceptual capability tuned to the actual differences in resource value can be observed. Overall, this appears to be solid work, with no obvious flaws in the math and reasonable results both of the analytical analysis and the numerical simulations. The manuscript is well written and should be comprehensible to interested readers. I did enjoy reading the paper and it yields some interesting results, as well as provides possible explanations for evolution of heterogeneous perceptual constraints, which could explain empirical deviations from the Ideal Free Distribution. In summary the manuscript, appears potentially suited for publication with PLoS Comp. Biol. However, I have a number of (minor) revision suggestion for the author:

- There are some open questions regarding the generality. The overall model assumes a simple linear dependency on the value difference, however it may be that there is e.g. a saturation of fitness benefits with increasing value (e.g. described by a logistic function). Would this change the qualitative picture?

- The model assumes strictly monotonously decreasing cost functions with delta, based on the mathematical expression in line 112. Below the definition of the cost function (Eq. 4) the author says that the results are general and other cost functions can be substituted. Further below, following Eq. 8, the authors it bases that results from c decreasing monotonically with delta.

It appears intuitively reasonable that the results

- In the discussion the author refers to some previous work without providing the reference at this point, as e.g. line 278-281. These should be provided.

should be general for strictly monotonously decreasing cost function. So convexity/concavity of the function does not play a role. This should be clearly stated at the latest following Eq. 4.

Here, however one can easily imagine that the cost function is not strictly monotonously decreasing but only monotonously decreasing: c(delta')<=c(delta) <-> delta'>delta, for example if at some point the difference is so large that distinguishing the patches does not incur any additional costs above a critical delta_crit. If this may change the results qualitatively, this should be clearly stated. The naive expectation would be simply the non-responsive simply become responsive with delta_crit. Thus as long as all deltas are below delta_crit then the results are the same (mathematically this last point is closely related to my first comment)

- The author refers to the non-responsive agents in the abstract, introduction, and discussion, as "free-riders". Classically free riding is defined by agents benefiting from some public good but not paying for it. Here the public good are the resources but the agents don't pay for access to resources but only for the ability to distinguish their quality. They just don't care about the value differences. In general, free-riding is also associated with the existence of social dilemmas, where the global (or group) optima differ from evolutionary stable strategies, with the letter leading to suboptimal reward at the group level. It is unclear on first glance whether this is the case here. Note, the scenario considered here is fundamentally different from the collective migration work by Guttal and Couzin cited in the manuscript. There, a fraction of agents evolves to not to detect the environmental gradient but only to be social, thus they free-ride on few informed agents that evolved gradient detection, with directional information being the public good generated by the informed agents. Thus, I am not sure this term is really justified here. Either the author should explain better why this is a form of free-riding or rewrite the corresponding parts.

- In the motivation there is strong emphasis on empirically observed under matching of IFD, which is largely based on the review of Kennedy and Gray, 1993. The author should have a look at: Milinski, M. (1994). Ideal free theory predicts more than only input matching: a critique of Kennedy and Gray's review. Oikos, 163-166.

The author correctly states that perceptual constraints have been suggested as an explanation for the under matching of the IFD. However, given this strong emphasis on this "flaw" of IFD, I feel the author should acknowledge the many other mechanisms that could be responsible for it (see e.g. references in Milinsky, 1994). Maybe even discuss briefly how the mechanisms discussed in this manuscript could be distinguished from these other mechanisms experimentally, i.e. how his theory could be tested.

- In the discussion the author refers to some previous work without providing the reference at this point, as e.g. line 278-281. These should be provided.

Reviewer #2: The manuscript by Mann et al is an interesting study on how evolution favours heterogeneous perceptual strategies in a foraging context. The article writing is lucid. It has analytical calculations whose conclusions are supported by agent-based simulations.

The author has done a really nice job of putting the study and its results in the context of existing body of work. The results by Mann in this manuscript are broadly consistent with the frameworks on producer scrounger dynamics as well as differential navigation strategies — as clearly explained by Mann. The author claims new and broadly applicable insights — especially on novel implications on ideal free distribution theory. Broadly speaking, if one requires a trait to utilise a resource and if that trait has some cost, for a wide-range of conditions, we find an evolution of differential strategies. In that sense, the results claimed by the author fall within the known framework of the evolution of differential strategies. Therefore, the novelty or newness of the results shown here are not clear to me.

What I found interesting was the exploration of how the strategies evolve in variable environmental context. Compared to the existing works on the evolution of heterogeneous strategies in a foraging context, I think the main new biological finding seems to be in this section on Variable environments. However, this section is not explored as much in terms of generality. Mann has modelled this variable environment by assuming that resource parameter r changes between two discrete values. Previous studies had only shown bimodal heterogenous strategies whereas this study shows that even trimodal strategies are possible; I think the author claims that it can be more than three if the variability of r happens between three values or more, but do not provide any analytical or numerical treatment for the same. Therefore, my questions are: Is this result general? What if there was continuous stochastic variation in resource variability parameter r? What if the number of discrete jumps between resources were more than two?

In summary: I think that the evolution of differential strategies in a foraging context are sort of known. It could be a much stronger if the focus is on the evolution of resource utilisation/extraction/finding strategies in the context of variable environments.

Minor point: Here is another such paper showing the evolution of differential strategies in a foraging/resource extraction context: Joshi, Jaideep, Åke Brännström, and Ulf Dieckmann. "Emergence of social inequality in the spatial harvesting of renewable public goods." PLoS computational biology 16.1 (2020): e1007483.

Reviewer #3: Attached.

**Have all data underlying the figures and results presented in the manuscript been provided?**

Reviewer #1: Yes

Reviewer #2: Yes

Reviewer #3: Yes

PLOS authors have the option to publish the peer review history of their article (what does this mean?). If published, this will include your full peer review and any attached files.

Reviewer #1: No

Reviewer #2: No

Reviewer #3: No
---

## [Decision Letter · Decision Letter 1]

11 Dec 2020

Dear Dr. Mann,

Thank you very much for submitting your manuscript "Evolution of heterogeneous perceptual limits and indifference in competitive foraging" for consideration at PLOS Computational Biology. As with all papers reviewed by the journal, your manuscript was reviewed by members of the editorial board and by several independent reviewers. The reviewers appreciated the attention to an important topic. Based on the reviews, we are likely to accept this manuscript for publication, providing that you modify the manuscript according to the review recommendations.

Thank you for revising your manuscript. The reviewers are positive about the revision but reviewer one points out a potential problem with the random switching term in the first equation. Please address this issue in your resubmission. Thank you.

Sincerely,

Jessica C. Flack

Associate Editor

PLOS Computational Biology

Stefano Allesina

Deputy Editor

PLOS Computational Biology

[LINK]

Thank you for revising your manuscript. The reviewers are positive about the revision but reviewer one points out a potential problem with the random switching term in the first equation. Please address this issue in your resubmission. Thank you.

Reviewer's Responses to Questions

**Comments to the Authors:**

Reviewer #1: I am mostly satisfied with the revisions made by the author, as he addressed/replied to all point raised by my previous review.

A repeated read of the revised manuscript, revealed however some issues with equation 1 which eluded me before and which the author should definitely address, as in its current form equation 1 is incorrect and very confusing:

1) The terms for random switching with rate gamma is missing in the equation. The following terms should be added:

-\\gamma \\rho(\\tau,A) + \\gamma \\rho(\\tau, B)

2) The notation for the different conditions is confusing. The "active" switching rate I is a scalar number but the notation with the brackets with (sign..) .implies its a function. Furthermore it appears twice as a product, which does not make sense. I guess, what the author means is that the different term are non-zero only if the difference of rewards is either smaller or larger then zero, or the threshold is below the difference. Here a mathematically correct and not confusing notation would be using the Heaviside function H(x) with \\Delta R for the reward condition and \\tau-\\Delta R as argument for the threshold condition. With this the RHS of the equation would read:

= [ \\rho(\\tau, B) \\times I \\times H( - \\Delta R) - \\rho(tau, A) \\times I \\times H(\\Delta R) ] \\times H( \\Delta R - \\tau) + missing random terms (see above)

I also suggest to use a multiply sign to better emphasize the different symbols, or (maybe better) use another symbol for the informed switching rate, instead of plain I. I would suggest using \\mathcal{I} or even just a simple \\beta.

Reviewer #2: Mann has addressed all the queries satisfactorily. I am looking forward to the publication of manuscript.

Reviewer #3: Typo in Figure S1: “ad panel B”

Otherwise, looks good.

**Have all data underlying the figures and results presented in the manuscript been provided?**

Reviewer #1: Yes

Reviewer #2: None

Reviewer #3: Yes

PLOS authors have the option to publish the peer review history of their article (what does this mean?). If published, this will include your full peer review and any attached files.

Reviewer #1: No

Reviewer #2: No

Reviewer #3: No
---

## [Editor Report · Decision Letter 2]

21 Jan 2021

Dear Dr. Mann,

We are pleased to inform you that your manuscript 'Evolution of heterogeneous perceptual limits and indifference in competitive foraging' has been provisionally accepted for publication in PLOS Computational Biology.

Best regards,

Jessica C. Flack

Associate Editor

PLOS Computational Biology

Stefano Allesina

Deputy Editor

PLOS Computational Biology

---

## [Editor Report · Acceptance letter]

5 Feb 2021

PCOMPBIOL-D-20-01455R2 

Evolution of heterogeneous perceptual limits and indifference in competitive foraging

Dear Dr Mann,

I am pleased to inform you that your manuscript has been formally accepted for publication in PLOS Computational Biology. Your manuscript is now with our production department and you will be notified of the publication date in due course.

With kind regards,

Alice Ellingham
